# Behavioral and neuronal underpinnings of safety in numbers in fruit flies

Clara H. Ferreira [1✉] & Marta A. Moita [1✉]

Living in a group allows individuals to decrease their defenses, enabling other beneficial behaviors such as foraging. The detection of a threat through social cues is widely reported, however, the safety cues that guide animals to break away from a defensive behavior and resume alternate activities remain elusive. Here we show that fruit flies display a graded decrease in freezing behavior, triggered by an inescapable threat, with increasing group sizes. Furthermore, flies use the cessation of movement of other flies as a cue of threat and its resumption as a cue of safety. Finally, we find that lobula columnar neurons, LC11, mediate the propensity for freezing flies to resume moving in response to the movement of others. By identifying visual motion cues, and the neurons involved in their processing, as the basis of a social safety cue this study brings new insights into the neuronal basis of safety in numbers.

---

[1] Champalimaud Research, Champalimaud Centre for the Unknown, 1400-038 Lisbon, Portugal. ✉email: clara.ferreira@neuro.fchampalimaud.org; marta.moita@neuro.fchampalimaud.org

Predation is thought to be a key factor driving group formation and social behavior (reviewed in ref. [1]). It has long been established that being in a group can constitute an anti-predatory strategy[2,3], as it affords the use of social cues to detect predators[4–7], enables coordinated defensive responses[8] or simply dilutes the probability of each individual to be predated[3]. A major consequence of this safety in numbers effect, reported in taxa throughout the animal kingdom, is that animals tend to decrease their individual vigilance[9], stress levels[10], or defensive behaviors[11] when in a social setting.

One of the most studied benefits of being in a group is the facilitated detection of behaviorally significant cues in the environment, as information about their presence can quickly spread across a large group of individuals[12]. In the context of threat detection, most research has focused on actively emitted signals, such as alarm calls and foot stamping (reviewed in refs. [13,14]). However, cues generated by movement patterns produced by defensive responses of surrounding prey can play a crucial role in predator detection. For example, crested pigeons use distinct wing whistles produced by conspecific escape flights[5] and rats use silence resulting from freezing, as alarm cues[4]. Recently, it has also been suggested that seismic waves produced by fast running in elephants promote vigilance in conspecifics[15]. This form of social detection of threat may be advantageous as it does not require the active production of a signal that may render the emitter more conspicuous and thus vulnerable. Although few studies demonstrated this phenomenon, it is described in distant vertebrate species.

Because living in a group allows individuals to decrease their defenses, it also enables other globally beneficial behaviors such as foraging. These selective forces on the evolution of social behavior have been demonstrated in a wide range of animals, from invertebrates to mammals[1,2]. Despite its wide prevalence, the mechanisms that lead to a decrease in defensive behaviors are largely unknown. Hence, in order to gain mechanistic insight into how increasing group size impacts defense behaviors, we decided to use *Drosophila melanogaster* since it allows the use of groups of varying size, the large number of replicates required for detailed behavioral analysis and genetic access to specific neuronal subtypes. Importantly, fruit flies display social behaviors in different contexts[16–21], namely social regulation of anti-predation strategies, such as the socially transmitted suppression of egg laying in the presence of predatory wasps[17] or the reduction in erratic turns during evasive flights when in a group, compared to when alone, in the presence of dragonflies[21].

In this study, we show that *Drosophila melanogaster* regulate their freezing behavior in response to threat as a function of group size. We identify the motion of others as a key regulator of freezing, with its cessation acting as a signal of danger and its presence constituting a safety signal. We further identify lobula columnar neurons 11 as major mediators of the usage of the movement of others as a safety cue. The identification of the sensory neurons responsible for social regulation of freezing opens up the possibility to gain mechanistic insight into the safety in numbers effect.

## Results

**Flies in groups display lower sustained freezing responses.** To simulate a predator's attack, we used a looming stimulus (Fig. 1a), an expanding dark disc, that mimics an object on collision course and elicits defense responses in visual animals, including humans (reviewed in refs. [22–24]). Individually tested fruit flies respond to looming stimuli with escapes in the form of jumps[25,26], in flight evasive maneuvers[27] or running as well as with freezing[28,29] when in an enclosed environment. In our setup, the presentation of

20 looming stimuli (Fig. 1a) elicited reliable freezing responses for flies tested individually and in groups of up to 10 individuals (Fig. 1b–e, Supplementary Fig. 1 shows that running and jumps are less prominent in these arenas). The fraction of flies freezing increased as the stimulation period progressed for flies tested individually and in groups of up to five flies; in groups of 6–10 individuals, the fraction of flies freezing only transiently increased with each looming stimulus (Fig. 1b). The fraction of flies freezing was maximal for individuals and minimal for groups of 6–10, while groups of 2–5 flies showed intermediate responses (Fig. 1b). The step-wise decrease between groups of five and six flies, does not seem to depend on fly density, as testing groups of five flies in a chamber that is 1-cm smaller, creating a density similar to that in groups of 7, did not impact freezing responses (Supplementary Fig. 2). At the level of each individual fly's behavior, flies tested alone spent more time freezing, 76.67%, interquartile range (IQR) 39.75–90.42%, during the stimulation period than flies in any of the groups tested (Fig. 1c; statistical comparisons in Supplementary Table 1). Flies in groups of 2–5 spent similar amounts of time freezing (for groups of 2: 31.67%, IQR 9.46–64.38% and for groups of 5: 43.08%, IQR 11.79–76.50%), while flies in groups of 6–10 displayed the lowest levels of freezing (for groups of 6: 8.08%, IQR 3.04–17.46% and for groups of 10: 3.33%, IQR 2–7.67%; Fig. 1c; statistical comparisons in Supplementary Table 1). The decrease in time spent freezing for flies tested in groups of 2–5, compared to individuals, was not due to a decrease in the probability of entering freezing after a looming stimulus (Fig. 1d; statistical comparisons in Supplementary Table 2), but rather to an increase in the probability of stopping freezing, i.e., resuming movement, before the following stimulus presentation (individually tested flies: $P(F_{exit}) = 0.08$, IQR 0–0.21, groups of 2: $P(F_{exit}) = 0.31$ IQR 0.11–0.78, groups of 5: $P(F_{exit}) = 0.54$ IQR 0.31–0.90; Fig. 1e; statistical comparisons in Table S3). Flies in groups of 6–10, were not only more likely to stop freezing (groups of 6: $P(F_{exit}) = 0.93$, IQR 0.80–1, groups of 10: $P(F_{exit}) = 1$, IQR 0.83–1; Fig. 1e; statistical comparisons in Table S3), but also less likely to enter freezing (groups of 6: $P(F_{entry}) = 0.35$, IQR 0.20–0.46, groups of 10: $P(F_{entry}) = 0.21$, IQR 0.10–0.36; Fig. 1c; statistical comparisons in Supplementary Table 2) compared to the other conditions. The decrease in persistent freezing with the increase in group size suggests that there is a signal conveyed by the other flies that increases in intensity with the increase in the number of flies tested together.

**Absence of movement promotes freezing.** We next examined whether flies respond to each other. We started by exploring the effect on freezing onset, as freezing has been shown to constitute an alarm cue in rodents, such that one rat freezing can lead another to freeze[4]. We decided to focus on groups of five flies, which showed intermediate freezing levels (Fig. 1). The onset of freezing both for individually tested flies and in groups of five occurred during and shortly after a looming stimulus (Fig. 2a). This window, of ~1 s, in principle allows for social modulation of freezing onset. Indeed, the probability of freezing onset at time $t$ gradually increased with increasing numbers of flies freezing at time $t-1$ (see Methods section), indicating that flies increase their propensity to freeze the more flies around them were freezing. This synchronization in freezing could result from flies being influenced by the other flies or simply time locking of freezing to the looming stimulus. To disambiguate between these possibilities we shuffled flies across groups, such that the virtual groups thus formed were composed of flies that were not together when exposed to looming. If the looming stimulus was the sole source of synchrony for freezing onset, then we should see a similar increase in probability of freezing by the focal fly with increasing

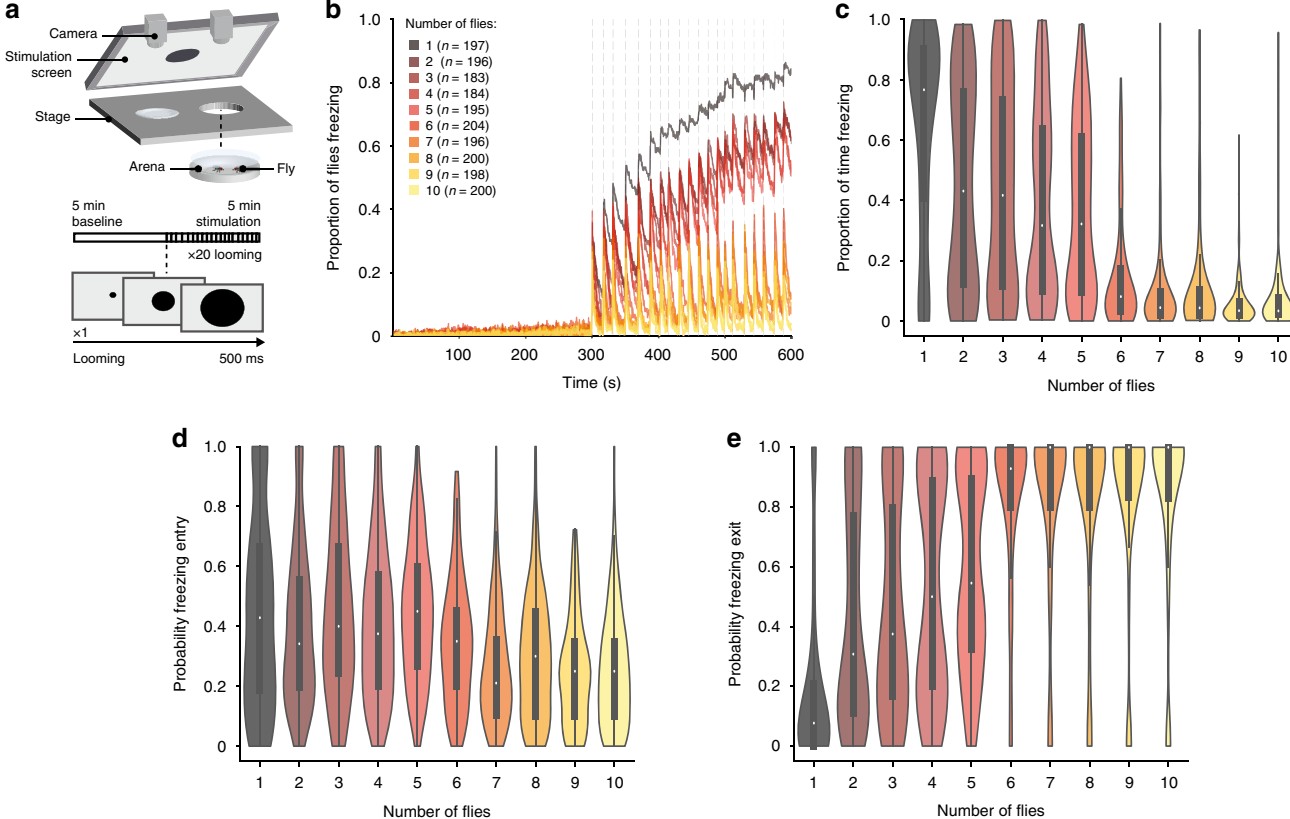

**Fig. 1 Analysis of the group effect on freezing responses. a** Experimental setup and protocol. We tested individuals and groups of up to 10 flies in backlit arenas imaged from above. After a 5-min baseline flies were exposed to twenty 500 ms looming presentations, every 10–20 s, indicated by vertical dashed lines. **b** Proportion of flies freezing throughout the experiment. **c–e** Violin plots representing the probability density of individual fly data bound to the range of possible values, with boxplots (elements: center line, median; box limits, upper (75) and lower (25) quartiles; and whiskers, 1.5× interquartile range). **c** Proportion of time spent freezing throughout the experiment. Statistical comparisons between conditions presented in Supplementary Table 1. **d** Probability of freezing entry after looming presentation. Statistical comparisons between conditions presented in Supplementary Table 2. **e** Probability of freezing exit before the following looming stimulus. Statistical comparisons between conditions are presented in Supplementary Table 3.

number of 'surrounding' flies freezing in the shuffled group. We found a weaker modulation of freezing onset by the number of flies freezing in randomly shuffled groups compared to that of the real groups of five flies (Fig. 2b; G-test, $g = 190.96$, $p < 0.0001$, df = 4). We corroborated this result by testing single flies surrounded by four fly-sized magnets whose speed and direction of circular movements we could control (Fig. 2c–f). During baseline, the magnets moved at the average walking speed of flies in our arenas, 12 mm per s, with short pauses as the direction of movement changed. Stopping the magnets upon the first looming stimulus and throughout the entire stimulation period led to increased time freezing (Fig. 2d) and increased probability of freezing entry upon looming (Fig. 2e), compared to all controls – individuals alone, magnets not moving throughout the entirety of the experiment and the exact same protocol (magnets moving during baseline then freezing) but in the absence of looming stimuli. The transition from motion to freezing is thus important, but not sufficient to drive freezing, since flies surrounded by magnets that do not move for the entire experiment froze to individually tested levels, but flies exposed to magnets that move and then freeze in the absence of looming stimuli did not freeze. Together these results suggest that flies use freezing by others as an alarm cue, which increases their propensity to freeze to an external threat, the looming stimulus.

**Movement of neighbors leads to freezing exit.** As the strongest effect observed across all group sizes was on freezing exit, i.e.,

the resumption of movement, we asked whether the propensity to exit freezing was also dependent on the number of surrounding flies that were freezing. To this end, we performed a similar analysis as for freezing onset and found that the higher the number of flies freezing, the lower the probability of the focal fly to exit from freezing. This effect was also decreased in shuffled groups (Fig. 3a; G-test, $g = 170.81$, $p < 0.0001$, df = 4). We then examined the contribution of mechanosensory signals in the decrease in freezing and found that collisions between flies play a minor role in the observed effect (Supplementary Fig. 3; statistical comparisons in Supplementary Tables 4–6), contrary to what happens with socially-mediated odor avoidance[16]. Next, we explored our intuition that motion cues from the other flies were the main players affecting exit from looming-triggered freezing. We formalized the motion cue (Fig. 3b), perceived by a focal fly, as the summed motion cues produced by the other four surrounding flies (we multiplied the speed of each fly by the angle on the retina, a function of the size of the fly and its distance to the focal fly, Fig. 3b). We then analyzed separately the summed motion cue perceived by focal flies during freezing bouts that terminated before the following looming stimulus (freezing with exit) and continuous freezing bouts (with no breaks in between looming stimuli; representative examples in Fig. 3b). Freezing bouts with exit had higher motion cue values (Fig. 3c) compared to continuous bouts ($p < 0.0001$, Freezing without exit = 0.64 IQR: 0.00–2.11, Freezing with exit = 2.79 IQR: 1.28–5.08).

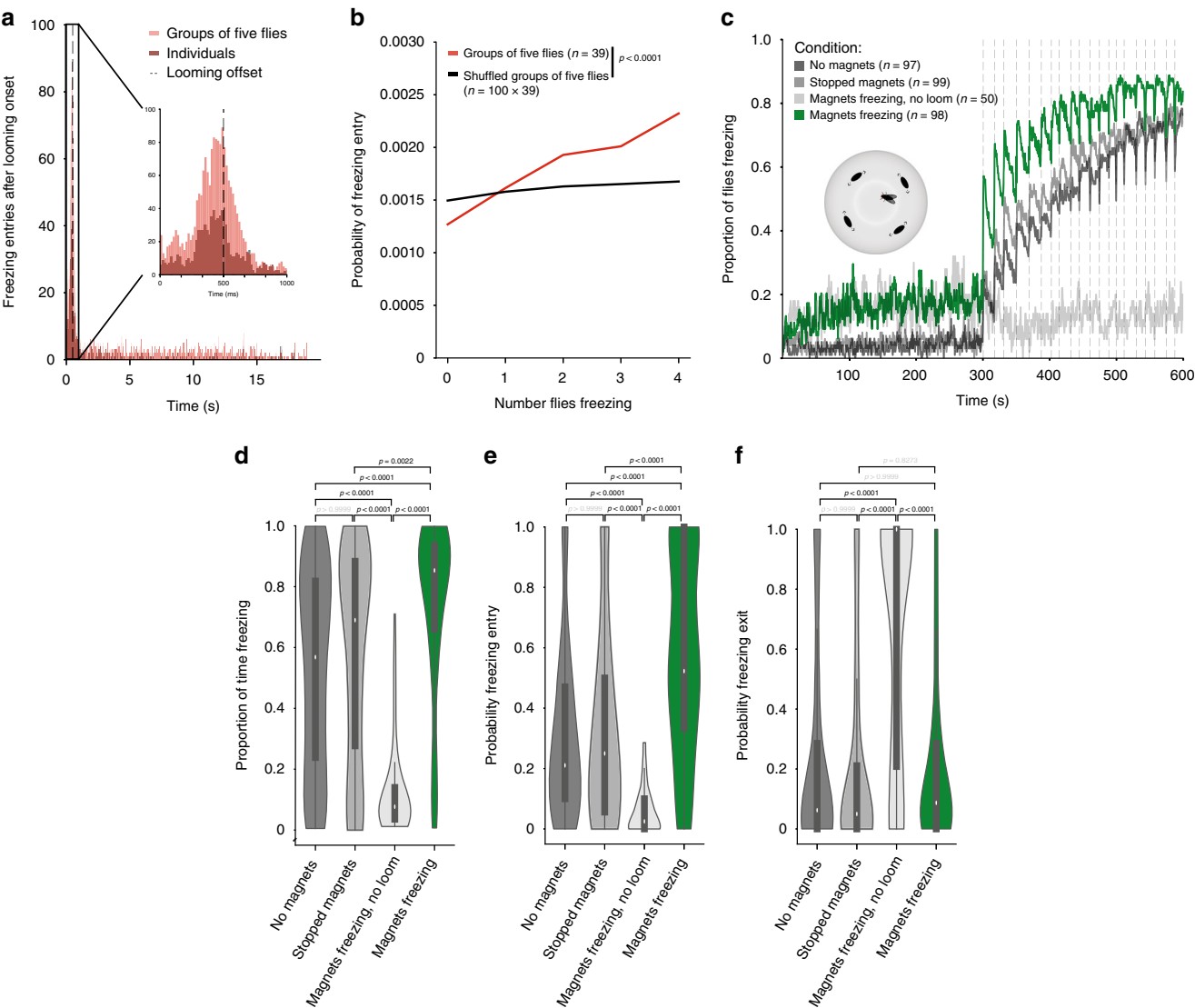

**Fig. 2 The group effect on individual freezing entry. a** Distribution of freezing entries after looming onset for flies tested individually and in groups of five. **b** Probability of freezing entry at time $t$ as a function of the number of other flies freezing at $t-1$ (see methods). P-value results from $\chi^2$ contingency test (G-test). **c–f** Simulating groups of five using movable magnets (stopped magnets – immobile magnets throughout the experiment; magnets freezing – magnets move during baseline becoming immobile magnets from the onset of the stimulation period). **c** Proportion of flies freezing throughout the experiment. **d–f** Violin plots representing the probability density of individual fly data bound to the range of possible values, with boxplots (elements: center line, median; box limits, upper (75) and lower (25) quartiles; whiskers, 1.5× interquartile range). **d** Proportion of time spent freezing throughout the experiment. **e** Probability of freezing entry after looming presentation. **f** Probability of freezing exit before the following looming stimulus. P-values result from Kruskal–Wallis statistical analysis followed by Dunn's multiple comparisons test.

We hypothesized that once flies start freezing, upon a looming stimulus, two processes determine whether a fly will exit freezing, resuming activity, or remain freezing: (1) an individual decision process, whereby flies make this binary decision irrespective of what the other flies are doing, possibly reflecting the number of looming stimuli the flies were exposed to and how much time has elapsed since the onset of freezing; (2) a social decision process whereby flies integrate the motion cues generated by their neighbors relying on this information to decide whether to stop freezing. To test this possibility, we modeled the decision to stay freezing or resume activity as a binary decision that follows a logistic function taking into account two parameters, the individual probability of exiting freezing before the next looming stimulus, and the motion cues of others (see Methods section). With this simple model we can predict whether a fly will stay freezing during the entire inter-looming interval or whether it

resumes activity in between looming stimuli, (area under the receiver operating characteristic curve AUROC = 0.87 ± 0.019, Fig. 3d). In addition, we found that the social cues explained a large fraction of the variance while individual behavior explains a small fraction (average variance explained by β-coefficient of social cues, $\beta_s = 0.85 \pm 0.019$, variance explained by β-coefficient for individual behavior $\beta_i = 0.15 \pm 0.019$, Fig. 3d, e).

To further test whether motion cues from others constitute a safety signal, we manipulated the motion cues perceived by the focal fly, while maintaining the number of flies in the group constant. An increase in the social motion cues, should enhance the group effect, and hence decrease the freezing responses of a focal fly. We compared groups of five wild-type flies with groups of one wild-type and four blind flies (norpA mutants; Fig. 4a). Blind flies do not perceive the looming stimulus and walk for the duration of the experiment; when a focal fly freezes surrounded

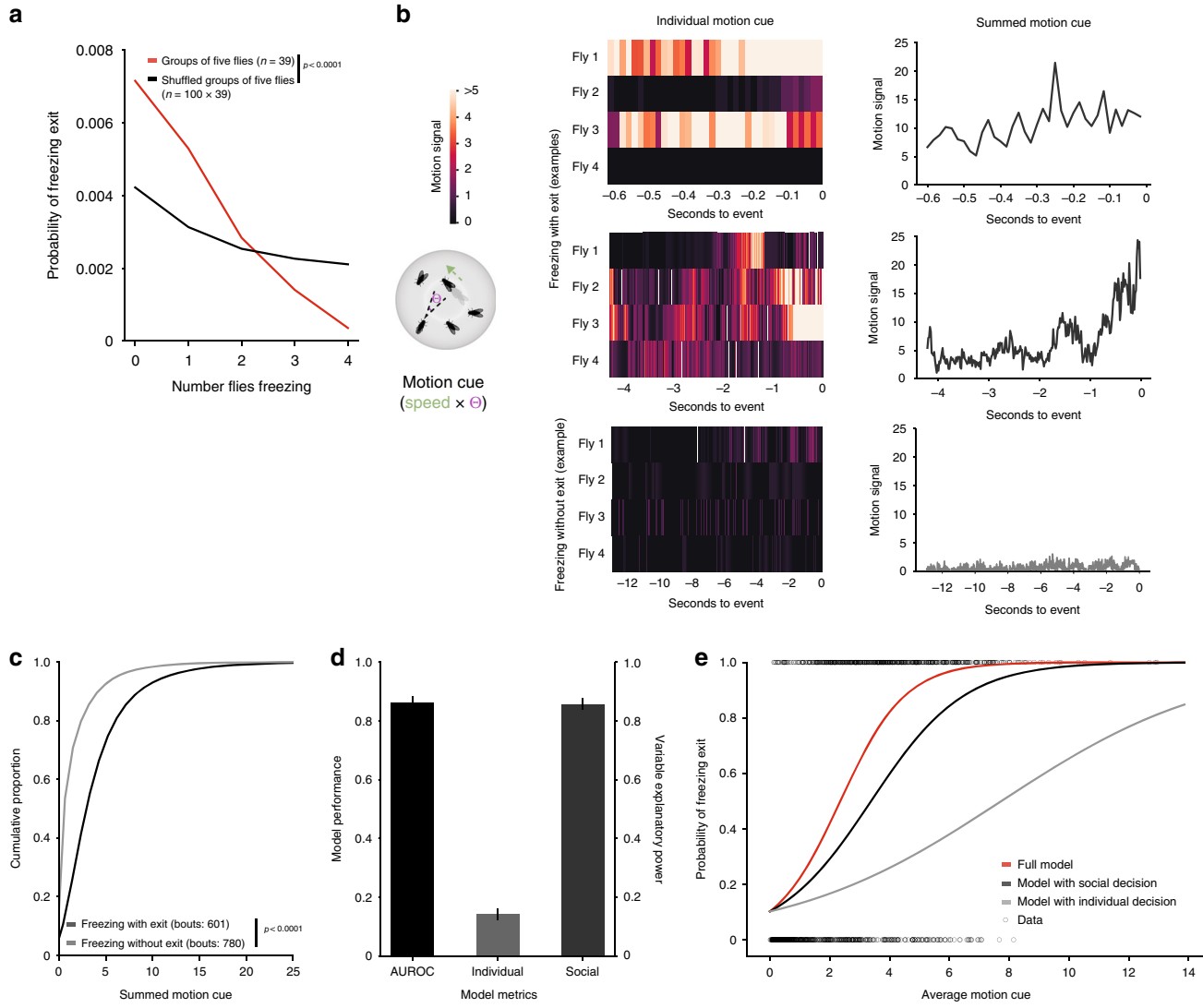

**Fig. 3 The group effect on individual freezing exit. a** Probability of freezing exit at time $t$ as a function of the number of flies freezing at time $t-1$ (see methods). P-value results from $\chi^2$ contingency test (G-test). **b** The motion cue is formalized as the other fly's speed multiplied by the angle ($\theta$) it produces on the retina of the focal fly (schematic). Representative examples of the motion cue starting in the 500 ms bin after looming offset for a focal fly until freezing exit or the end of the inter-looming interval (without freezing exit): heatmaps show the individual motion cues for each of the four surrounding flies and the line graphs show the summed motion cue of these four flies. **c** Cumulative distributions of the summed motion cues ($x$ axis cut at motion cue = 25). P-value results from Kolmogorov–Smirnov test. **d, e** Logistic regression model of the decision to stop or continue freezing as a function of individual and social decision processes (10,000 bootstrapping events). **d** Mean and standard deviation of model performance (AUROC – area under the receiver operating characteristic curve, black) and explanatory power of the individual (light gray) and social processes (dark gray). **e** Binary freezing data and average logistic predictive probabilities using individual and social coefficients alone or combined as a function of the average motion cue.

by four blind flies it is thus exposed to a higher motion cue during the stimulation period than a focal fly in a group of five wild-type flies (Fig. 4a). When surrounded by blind flies, the fraction of focal flies freezing throughout the stimulation period was lower than the fraction of flies freezing in a group of wild-type flies (Fig. 4b). Further, the increase in motion cues in groups with blind flies decreased the amount of time a fly froze compared to that of groups of wild-type flies (6.17% IQR 2.17–15.25% versus 19.58% IQR 8.20–57.12; $p < 0.0001$; Fig. 4c). This reduction in freezing resulted mostly from a decreased probability of freezing entry (wild-type groups: $P(F_{entry}) = 2.57$ IQR 0.15–0.39, groups with blind flies: $P(F_{entry}) = 0.49$ IQR 0.25–0.61, $p < 0.0001$; Fig. 4d) and slightly increased probability of exiting freezing (wild-type groups: $P(F_{exit}) = 0.83$ IQR 0.39–1, groups with blind flies: $P(F_{exit}) = 0.89$ IQR 0.71–1; Fig. 4e). Hence, a focal fly surrounded

by four blind flies behaves similarly to flies in groups of more than six individuals. Importantly, the decrease in persistent freezing was not due to an increased role of collisions on freezing breaks (Supplementary Fig. 4). We further tested whether any type of visual signal could alter individual freezing in the same manner as the motion cues generated by flies in the group, by presenting a visual stimulus with randomly appearing black dots with the same change of luminance as the looming stimulus but without motion (used as control stimulus in our previous study[28]) 4.5 s after each looming presentation. This stimulus, which could work as a distractor, did not alter the proportion of time freezing nor the probability of freezing entry or exit (Supplementary Fig. 5). Finally, we also assessed the role of other sensory cues, namely olfaction and gustation. Using near-anosmic mutants and testing the impact of contacts, required for gustatory cues, on the

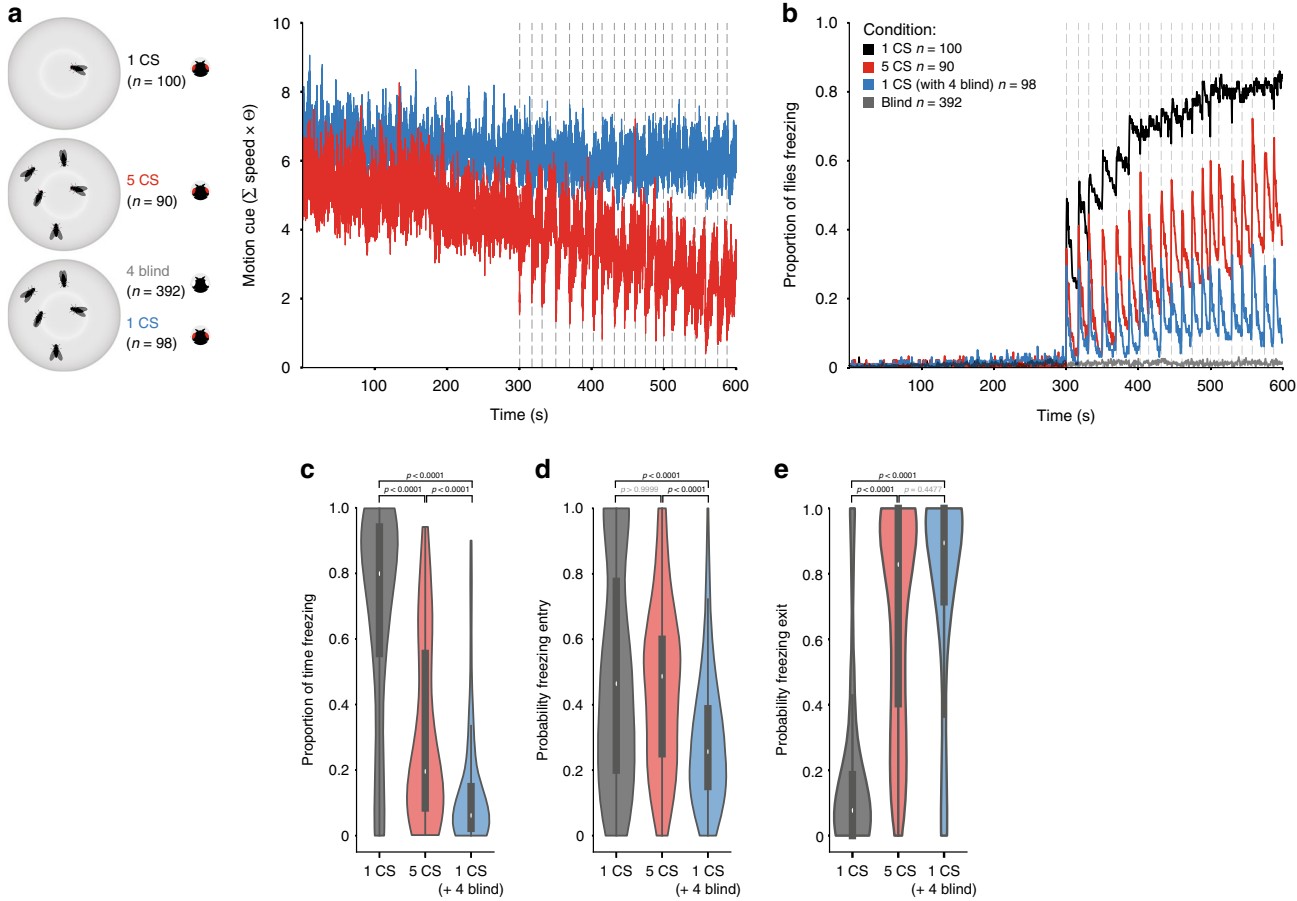

**Fig. 4 Manipulating the motion cue in groups of five flies. a** The summed motion cue of surrounding flies for groups of five wild-type flies and groups with one wild-type and four blind flies. **b** Proportion of flies freezing throughout the experiment. **c–e** Violin plots representing the probability density of individual fly data bound to the range of possible values, with boxplots (elements: center line, median; box limits, upper (75) and lower (25) quartiles; and whiskers, 1.5× interquartile range). **c** Proportion of time spent freezing throughout the experiment. **d** Probability of freezing entry after looming presentation. **e** Probability of freezing exit before the following looming stimulus. *P*-values result from Kruskal–Wallis statistical analysis followed by Dunn's multiple comparisons test.

logistic regression model we found that olfaction and gustation are unlikely to play a role in the group response (Supplementary Fig. 6).

Together, these results show that flies use motion cues generated by their neighbors to decide whether to stay or exit freezing, raising the possibility that motion cues produced by others could constitute a safety signal leading flies to resume activity.

**Lobula columnar neurons 11 mediate group effect**. Having identified motion cues of others as the leading source of the group effect on freezing, we decided to test the role of visual projection neurons responsive to the movement of small objects. In particular, lobula columnar 11 (LC11)[30,31] neurons have been shown to respond to moving objects of angular sizes[31] that could be generated by moving flies within our arenas. Furthermore, the behavioral relevance of these neurons was as yet unidentified. To silence LC11 neurons we used one fly line, an *LC11-GAL4*[31], to drive the expression of either Kir 2.1[32], a potassium channel that hyperpolarizes neurons decreasing their ability to fire action potentials, or tetanus toxin light chain (TNT), which cleaves neuronal synaptobrevin preventing synaptic release of neurotransmitter[33]. Constitutively silencing LC11 neurons did not alter looming-triggered freezing of flies tested individually (Supplementary Fig. 7). Conversely,

for LC11-silenced flies tested in groups of five, the fraction of flies freezing increased throughout the experiment (Fig. 5a). Moreover, experimental flies in groups of five froze longer (~3.5-fold increase for *LC11-GAL4>Kir2.1*, and ~2-fold increase for *LC11-GAL4> (+) TNT* compared to controls; Fig. 5b), which was not due to an increase in the probability of freezing entry (Fig. 5c), but rather to a decrease in the probability of freezing exit (Fig. 5d; *LC11-GAL4>Kir2.1* 0.077 IQR 0.00-0.17 and *Empty-GAL4>Kir2.1* 0.59 IQR 0.15-1; *LC11-GAL4>(+)TNT* 0.17 IQR 0.06–0.50, and *LC11-GAL4>(−)TNT* 0.33 IQR 0.14–0.77). These data, together with the identification of visual motion cues as mediators of group freezing responses, point to the role of LC11 neurons in this process. However, given that *LC11-GAL4*, despite its sparseness, also directs expression outside these neurons, namely in the descending neurons DNg26[34], we cannot at this moment fully rule out the effect of expression outside LC11. In addition, the observed effect of silencing neurons targeted by the LC11-GAL4 line on freezing in groups may be adult specific or due to developmental effects. Finally, to assert the specificity of our manipulation we expressed Kir2.1 in another LC neuron class, LC20[30], which are not known to respond to small moving objects, and found that it does not alter group behavior (Supplementary Fig. 8). In summary, silencing LC11 neurons renders flies less sensitive to the motion of others, specifically decreasing its use as a safety cue that downregulates freezing.

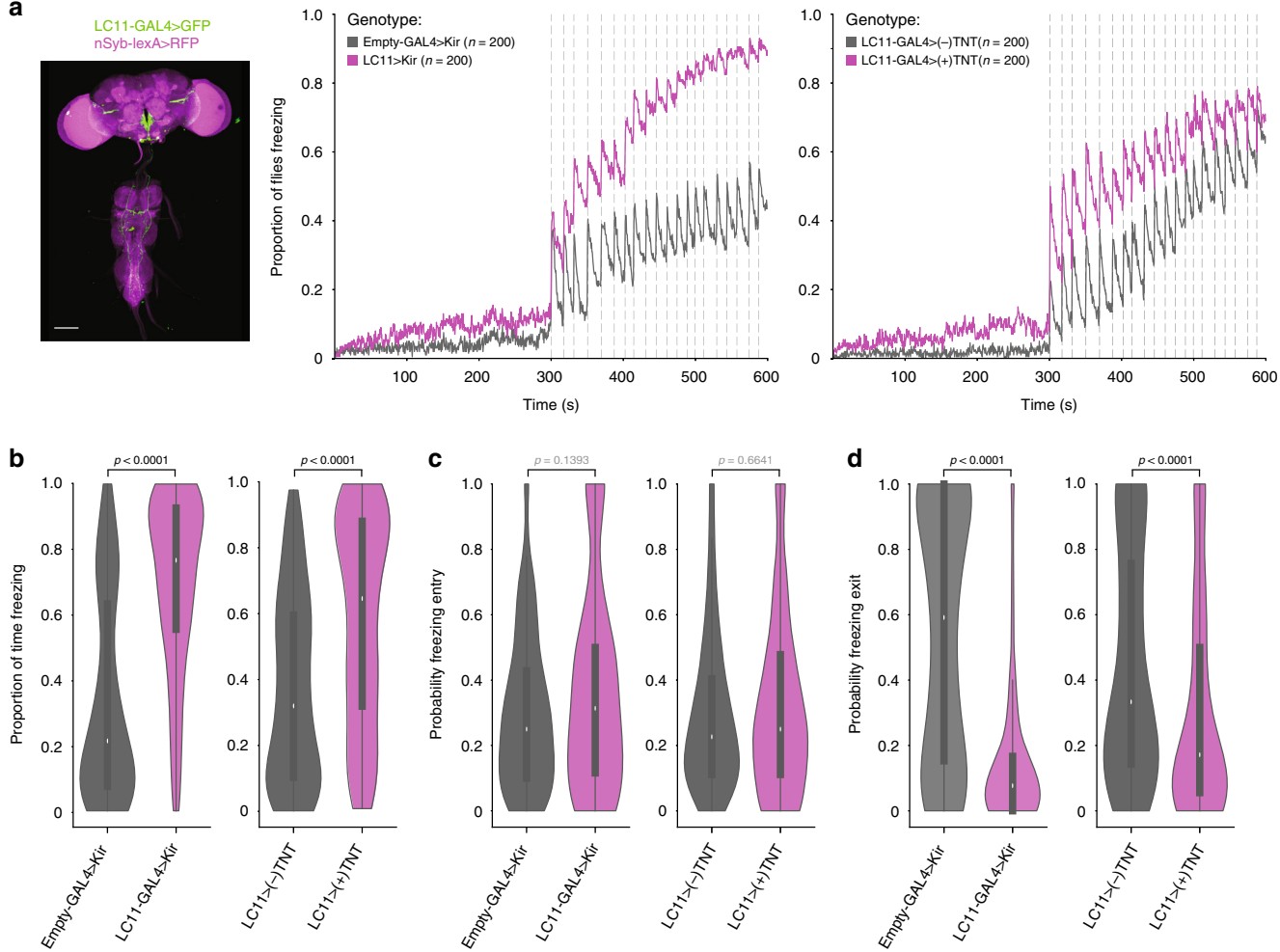

**Fig. 5 Manipulating lobula columnar neurons 11 (LC11). a** Anatomy (scale bar, 100 μm) and proportion of flies freezing throughout the experiment in groups of five, for *LC11-GAL4 > Kir2.1* and *LC11-GAL4 > (+)TNT* depicted in purple and controls (gray). **b**–**d** Violin plots representing the probability density distribution of individual fly data bound to the range of possible values, with boxplots (elements: center line, median; box limits, upper (75) and lower (25) quartiles; and whiskers, 1.5× interquartile range). **b** Proportion of time spent freezing throughout the experiment. **c** Probability of freezing entry after looming presentation. **d** Probability of freezing exit before the following looming stimulus. *P*-values result from two-tailed Mann–Whitney test.

## Discussion

In this study, we show that flies in groups display a reduction in freezing responses that scales with group size. Detailed behavioral analysis and quantitative modeling together with behavioral and genetic manipulations, allowed us to identify freezing as a sign of danger and activity as a safety cue. These findings are consistent with the hypothesis that safety in numbers may partially be explained by the use of information provided by the behavior of others. Moreover, we show that visual projection LC11 neurons are involved in processing motion cues of others to downregulate freezing.

With the experiments reported here, we extend to invertebrates the notion of defensive behaviors, in this case freezing, as alarm cues. In addition, freezing may constitute a public cue that can be used by any surrounding animal regardless of species. Indeed, we show that freezing by dummy flies enhances freezing in response to looming stimuli.

Importantly, we also identify a social cue of safety. In our paradigm, flies responded to the threatening looming stimulus with freezing. At some point after the stimulus, flies can exit freezing resuming movement, until a new looming stimulus is presented, triggering freezing again. The more stimuli the flies were exposed to the less likely they are to exit freezing before

the next looming. This pattern suggests that the resumption of activity reflects the level of safety, such that when in groups the movement of others can constitute a cue of safety leading to further activity. Using a logistic regression model and manipulating the levels of movement by neighboring flies we demonstrated that motion cues of others strongly determine the propensity of flies to resume activity. In a prior study[4] we showed that when we present an auditory cue of movement to rats that are freezing in response to the display of freezing by another rat, they resume activity. Although in line with the present findings, we did not explicitly test whether this motion cue constituted a safety cue, as we have done here.

While there are known examples of the use of auditory motion cues to infer the presence or absence of a threat in vertebrate species, here we show that flies use visual motion cues. This may relate to the fact that *Drosophila melanogaster* use short range auditory signals, whereas visual cues can be detected at larger distances. Silencing LC11 neurons, which process visual information, responding to motion of small visual objects, disrupted the use of motion cues from neighboring flies as a safety cue. Though motion also generates vibrations cues and these can be used to detect the movement of other flies[35], our results suggest visual cues play a

predominant role. Furthermore, other LC neurons have been implicated in processing visual stimuli in social contexts, namely *fru* + LC10a important for the ability of males to follow the female during courtship[36]. LC cells in the fly seem to be tuned for distinct visual features, and activating specific LC cells leads to distinct approach or defensive responses[30]. It will be interesting to study to what extent there is specificity or overlap in visual projection neurons for behaviors triggered by the motion of others. The parallels between visual systems of flies and humans (reviewed in refs. [37,38]), despite the lack of any common ancestor with an image forming visual system, suggest that shared mechanisms underlying visuomotor transformations represent general solutions to common problems that all organisms face individually or as a group.

Motion plays a crucial role in predator-prey interactions. Predator and prey both use motion cues to detect each other using these to make decisions about when and how to strike or whether and how to escape[39–42]. Furthermore, prey animals also use motion cues from other prey as an indirect cue of a predator's presence[4,12,43]. We believe that the current study opens a new path to study how animals in groups integrate motion cues generated by predators, their own movement, and that of others to select the appropriate defensive responses.

## Methods

**Fly lines and husbandry.** Flies were kept at 25 °C and 70% humidity in a 12 h:12 h dark:light cycle. Experimental animals were mated females, tested only once when 4–6 days old.

Wild-type flies used were Canton-S. *LC11-GAL4 w[1118]; P{y[+t7.7] w[+mC] = GMR22H02-GAL4}attP2, LC20-splitGAL4 w[1118]; P{y[+t7.7] w[+mC] = R35B06-GAL4.DBD}attP2 PBac{y[+mDint2] w[+mC] = R17A04-p65.AD}VK00027* and *w[*] norpA*[36] were obtained from the Bloomington stock center. *10XUAS-IVS-eGFPKir2.1 (attP2)* flies were obtained from the Card laboratory at Janelia farm. *UAS-CD8::GFP; lexAop-rCD2::RFP*[44] recombined with *nSyb-lexA.DBD::QF.AD* (obtained from the Bloomington stock center) were obtained from Wolf Huetteroth, University Leipzig. *UAS-(+) TNT and UAS-(−) TNT*[33] were obtained from the Chiappe lab, Champalimaud Research. The olfactory mutant *IR8a1; IR25a2; GR63a1, ORCO1* were obtained from the Benton lab, University of Lausanne.

**Behavioral apparatus and visual stimulation.** We imaged unrestrained flies in 5 mm thick, 11° slanted polyacetal arenas with 68 mm diameter (central flat portion diameter 32 mm). Flies were not restricted to the arena floor, as during initial experiments we observed no difference in defensive responses for flies on the floor or ceiling. Visual stimulation (20 500-ms looming stimuli, a black circle in a white background, with a virtual object length of 10 mm and speed 25 cm per s (l/v value of 40 ms) as in ref. [28]) was presented on an Asus monitor running at 144 Hz, tilted 45° over the stage (Fig. 1a). For the experiments with random dots, 4.5 s after the looming presentation we presented a visual stimulus consisting of appearing black dots at random locations on the screen to reach the same change in luminance as the looming stimulus[28].

The stage contained two arenas, backlit by a custom-built infrared (850 nm) LED array. Videos were obtained using two USB3 cameras (PointGrey Flea3) with an 850-nm-long pass filter, one for each arena.

For the experiments with the magnets (Fig. 2), we used an electromechanical device developed by the Scientific Hardware Platform at the Champalimaud Centre for the Unknown. It consists of an adapted setup in which a rotating transparent disc with five incorporated neodymium magnets moves under the arena. A circular movement is induced by an electric DC gearhead motor transmitted via a belt to the disc. This allows magnetic material placed on the arena to move around in synchronized motion. The motor is controlled by a custom-made electronic device, connected to the computer, through a dedicated Champalimaud Hardware Platform-developed software. For the experiments of freezing magnets during stimulation, with or without stimulus, the magnets rotated at 12 mm per s with a change in direction every 50 s during the baseline; as soon as the stimulation period started, in synchrony with the first looming stimulus, the magnets ceased movement, until the end of the experiment.

**Video acquisition and analysis.** Videos were acquired using Bonsai[45] at 60 Hz and 1280 width × 960 height resolution. We used IdTracker[46] to obtain the position throughout the video of each individual fly. The video and the IdTracker trajectories file were then fed to the 'Fly motion quantifier', developed by the

Scientific Software Platform at the Champalimaud Centre for the Unknown in order to obtain the final csv file containing not only position and speed for each fly, but also pixel change in a region of interest (ROI) around each fly, defined by a circle with a 30 pixel radius around the center of mass of the fly.

**Data analysis.** Data were analyzed using custom scripts in spyder (python 3.5). Statistical testing was done in GraphPad Prism 7.03, and non-parametric, Kruskal–Wallis followed by Dunn's multiple comparison test or two-tailed Mann–Whitney tests were chosen, as data were not normally distributed (Shapiro–Wilk test). Probabilities were compared using the $\chi^2$ contingency test in python (G-test).

Freezing was classified as 500 ms periods with a median pixel change over that time period <30 pixels within the ROI. The proportion of time spent freezing was quantified as the proportion of 500 ms bins during which the fly was freezing.

We calculated the proportion of freezing entries upon looming and exits between looming stimuli (Fig. 1) using the following definitions: (1) freezing entries corresponded to events where the fly was not freezing before the looming stimulus (a 1-s time window was used) and was freezing in the first 500-ms bin after the looming stimulus; (2) freezing exits were only considered if sustained, that is, when the fly froze upon looming but exited from freezing and was still moving by the time the next looming occurred, i.e., the first 500-ms bin after looming the fly was freezing and in the last 500-ms bin before the next looming the fly was not freezing.

To determine the time of freezing onset or offset (Figs. 2a, b and 3a), we used a rolling window of pixel change (500-ms bins sliding frame by frame) and the same criterion for a freezing bin as above. Time stamps of freezing onset and offset were used to calculate the probability of entering and exiting freezing as a function of the number of flies freezing. For freezing entries after looming as well as probabilities of entering and exiting freezing, we considered only instances in which the preceding 500-ms bin was either fully non-freezing or freezing. To determine the numbers of others freezing at freezing entry or exit we used a 10 frame bin preceding the freezing onset or offset timestamp.

Distances between the center of mass of each fly were calculated using the formula $\sqrt{(x2-x1)^2+(y2-y1)^2}$, and we considered a collision had taken place when the flies reached a distance of 25 pixels. The motion cue was determined as $\sum \text{speed} \times \text{angle on the retina}(\theta)$ where $\theta = 2 \arctan\left(\frac{\text{size}}{2 \times \text{distance}}\right)$.

To analyze the motion cue for freezing bouts with and without exit (Fig. 3b, c), we defined freezing bouts with exit as bouts where flies were freezing in the 500 ms following the looming stimulus offset and resumed moving before the next looming stimulus (up until the last 500 ms before the looming stimulus onset) and freezing bouts without exit as those where freezing persisted until the next looming. Cumulative proportions of motion cues for freezing with and without exit were compared using the Kolmogorov–Smirnov test.

To model the decision to stay frozen or resume movement we used the scikit-learn logistic regression model. Briefly, we analyzed freezing behavior in between looming stimuli, categorizing freezing bouts into two types: freezing bouts that ended with an exit before the next looming (to which we assigned a value of 1), and continuous freezing bouts, without an exit until the next looming (value of 0). We used freezing bout type as the dependent variable. The independent variables were the probability of an individual fly exiting from freezing within the same inter-looming interval (calculated from the data of flies tested individually) ($V_i$); and the sum of the motion cue generated by neighboring flies, divided by the bout length ($V_s$). We performed a K-fold cross-validation with four splits and used 10,000 times bootstrapped data with replacement. To determine the explanatory power of each predictor, we determined the associated fraction of variance using the following formula (shown for variable $V_i$): $\frac{\sum V_i \beta_i}{\sum V_i \beta_i + \sum V_s \beta_s}$, where $\beta_s$ and $\beta_i$ are, respectively, the $\beta$-coefficients of social cues and individual behavior.

**Imaging.** *LC11-GAL4>UAS-CD8::GFP; nSyb-lexA>lexAop-rCD2::RFP* and *LC20-splitGAL4>UAS-CD8::GFP; nSyb-lexA>lexAop-rCD2::RFP* 3-day-old females were processed for native fluorescence imaging as in ref. [47]. In brief, brain were dissected in ice-cold 4% PFA and post-fixed in 4% PFA for 40–50 min. After 3 × 20 min washes with PBST (0.01 M PBS with 0.5% TritonX) and 2 × 20 min washes in PBS (0.01 M) brains were embedded in Vectashield and imaged with a ×16 oil immersion lens on a Zeiss LSM 800 confocal microscope.

**Reporting summary.** Further information on research design is available in the Nature Research Reporting Summary linked to this article.

## Data availability

All raw data files are available at https://doi.org/10.6084/m9.figshare.12554663. Source data are provided with this paper.

## Code availability

Code available upon request.

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

## Acknowledgements

We would like to thank: the Scientific Software Platform at the Champalimaud Centre for Unknown for developing the Fly motion quantifier; the Scientific Hardware platform for developing the magnet setup; Wolf Huetteroth (University of Leipzig) for help with imaging fly lines; Ricardo Vieira for help streamlining the video analysis pipeline; Gil Costa for the illustrations in Figs. 1a, 3b, and 4a; the Moita lab, particularly Anna Hobbiss and Ricardo Neto, as well as Eugenia Chiappe and Gonzalo de Polavieja for fruitful discussions and comments on the manuscript; Alfonso Renart and João Afonso for help with the logistic regression model; and Rui Gonçalves for invaluable fly pushing technical assistance during the revision process. This work was supported by Fundação Champalimaud, ERCStG337747-CoCO and ERCCoG819630-A-Fro.

## Author contributions

C.H.F. performed all experiments and analyzed the data. C.H.F. and M.A.M. designed the experiments, discussed results, and wrote the manuscript.

## Competing interests

The authors declare no competing interests.
