## [Peer Review File · Nature Communications]

Reviewers' Comments:

Reviewer #1:

Remarks to the Author:

Ferreira and Moita present a behavioral and neurogenetic analysis of the impact of group size in *Drosophila* on the probability that an individual animal will exhibit a defensive freezing behavior in response to a threatening (looming) visual cue. The authors demonstrate that when in larger groups (i.e., higher density), individual flies spend less time in a freezing state in response to such a cue, because of higher probability of exiting this state. They provide evidence that this is due to detection of motion of other flies in the group, mediated, at least in part, by the LC11 visual projection neurons (previously shown to respond physiologically to small moving objects).

This is an original, fascinating, rigorously analyzed and clearly presented study. The results will be of general interest because of its wider implications for the social influences on individual animal behavior, and the potential of future exploitation of this tractable system to understand the neurobiological underpinnings of this phenomenon.

I have only a few minor comments:

Figure 1B: the grouping of behavioral responses into three categories (i.e., 1 fly, 2-5 flies and 6-10 flies), as opposed to a linear impact of group size (animal density) is intriguing. Can the authors comment on what might define the apparently threshold between groups of 5 flies and groups of 6 flies?

Line 90: typo in "were"

Figure 2A: I don't understand the color key; in the plot there are orange bars and dark orange bars, but the key indicates orange, black and grey.

Figure 5B: the Kir2.1 control parental line show higher freezing behavior than in panel A (and the control Gal4 line). Is this just a fluke of this set of experiments?

The use of auditory cues as defense signals in other species made me wonder whether this sensory modality contributes to social regulation of freezing behavior in flies. (As a precedent, analysis of courtship has shown a male responds to auditory cues from the footsteps of females (PMID: 18802468)). While visual cues are most likely to be the most important signal, can the authors rule in or out other modalities? (No experiments are needed, but they could comment on this in the Discussion).

Discussion: this feels a bit long - the conclusions and broader implications of the work would come across stronger with a bit of pruning. For example, lines 216-231 are more background material (which might fit better in the (currently brief) introduction), and in lines 257-267, the highlighting of potential parallels between fly and visual processing feels a bit labored (and not really key for this paper).

Line 320: the web-link given is uninformative (I was expecting to have access the software).

Reviewer #2:

Remarks to the Author:

In the manuscript by Ferreira and Moita, authors investigate how freezing behavior in response to looming stimuli is regulated in groups using *Drosophila melanogaster* as a model organism. They first show that freezing response of flies decrease with increase in group size. This decrease is not a general unresponsiveness to a looming stimulus, but due to an increase in the probability of stopping freezing. Using fly-sized magnets, authors show when objects surrounding a fly transition from moving to freezing, that individual fly is more likely to freeze as well. Based on their results, authors hypothesized that motion cues from other flies or surrounding objects might mediate exit from looming-triggered freezing. To test this hypothesis, they formalized the motion signals perceived by a focal fly using three parameters; speed, size, and distance. Then, they showed that focal flies which had higher motion cue values were more likely to resume movement. To further examine the effect of motion cues on freezing, they manipulated the motion cues in the focal fly by using blind flies, that are insensitive to the looming stimulus and therefore do not change walking behavior during stimulus presentation. They found that the amount of time a fly froze decreased when surrounded by blind flies, compared to that of groups of wild-types, suggesting flies at least in part use motion cues of others to decide whether to stay frozen or exit freezing. Finally, they found that a group of visual projection neurons, LC11, mediate group effect on freezing responses; silencing of LC11 neurons make flies less sensitive to the motion of others.

Overall, I enjoyed reading the manuscript and I find the results interesting. This paper shows that flies in groups display a reduction in freezing responses that scales with the group size. However, I am not certain the reduction in freezing in groups is due to only motion sensing. There might be proprioceptive or chemosensory cues that are involved in the inhibition of looming responses that are not addressed in the manuscript.

I have major and minor points indicated below.

Major points:

1. In figure 1, authors showed that flies in groups of 6 to 10 were less likely to enter to a freezing state compared to individuals and groups of 2 to 5 flies. The difference between groups of 2-5 and groups of 6-10 is a step wise increase rather than a gradual difference. Is this an artifact of the chamber size or is this a biologically relevant result? If authors increase the size of the chamber would they expect different results in their assays?
2. In supplementary figure 2, authors investigate whether mechanosensory interactions contribute to freezing behavior and found minor impact of collisions on freezing behavior. Based on these preliminary tests, authors conclude that freezing behavior is not related to mechanosensation. I think these experiments are a good start however they need to be expanded to exclude the possibility that any proprioceptive feedback is not involved in this behavior. Authors also ignore, any chemosensory signal that can impact this behavior. In many species, alarm pheromones play a key role in directing group behavior to predators or threats in the environment. These experiments are simple, fly mutants are available to test impact of olfactory or taste signal on freezing behavior in fly groups. I think authors either need to conduct these experiments or explain in the text why they exclude the possibility that chemosensation or proprioception can regulate freezing behavior.
3. In figure 5, authors showed that inhibiting LC11 neurons by Kir2.1 expression reduces group effect on looming-triggered freezing behavior and they argue that LC11 neurons are involved in processing motion cues of other flies to regulate freezing response. In these experiments Kir2.1 is expressed throughout the development and LC11 neurons have been constantly hyperpolarized. These experiments need to be repeated by silencing LC11 neurons only in adults using temporal inactivation of these neurons with available reagents such as temperature controlled shibire. In addition, they can also activate LC11 neurons and examine whether freezing behavior will be suppressed in individual flies when LC11 neurons are active during looming stimulation. If LC11 neurons indeed mediate

freezing responses in grouped flies, activation of these neurons should induce the opposite phenotype caused by neural silencing.

4. In figure 5B, there is a significant difference between two control groups in freezing behavior; LC11 split-GAL4 and UAS-Kir2.1. What is the source of this variation in freezing behavior? Did they observe any behavioral or genetic problems in the LC11 split-GAL4 strain? Please clarify.

Minor points:

1. Please revise the gray color scheme in Figure 2. It is very hard to distinguish different experimental groups especially in Figure 2C. Also, please indicate the control groups in the figures more clearly. Stopped magnets and freezing magnets sounds too similar and confusing for the reader.

2. In the line 279, authors mentioned that they used mated females for their experiments. Mating could influence various aspects of animal behaviors including foraging, egg laying, courtship, and receptivity. Would they expect differences in freezing behavior if virgin females or males are used?

3. Authors use blind flies in the Figure 4, but they do not describe the genotype or the strain used in the methods. Please indicate which blind fly strain is used in the paper.

Reviewer #3:

Remarks to the Author:

Drs. Ferreira and Moita Show, for the first time, changes in defensive behaviour of fruit flies that are depending on group size. Authors used a looming stimulus that would elicit escape jumps in open space and freezing in the enclosed environment used here. Flies were tested individually or in groups of 2 to 10 flies. Individual flies froze most often, groups of 6 to 10 flies least often, groups of 2-5 flies were intermediate. Freezing occurs within a 1s time window after stimulus onset and peaks around 0.5s. It is argued that this is enough time for taking freezing of other individuals into account. The interdependence is statistically shown and elegantly demonstrated in a setup with one real and four dummy flies that could be moved and stopped (frozen) at will. The higher the summed retinal speed caused by the other flies, the less likely a test fly will enter freezing, and the more likely a frozen test fly will give up freezing. This is nicely demonstrated on a seeing fly with four blind companions, who do not freeze upon looming. Finally, lobula columnar neurons LC11 were inactivated using neurogenetic tools. This measure does not affect the reaction of individuals' freezing upon looming stimuli. However, it reduces the probability of exiting freezing. Authors conclude that LC11 neurons are involved in the perception of the motion of companion flies. Overall, the findings are novel, substantiated by data and will find the interest of a large community.

Major

Fig.3 B right side x-y-plots. Y-axis labels are missing and plots are not explained in the legend. What is the difference (despite x-axis range)? Shouldn't the 3rd contain the 2nd from -4s to 0s and the 2nd the first from -0.6s to 0s?. If this is "Summed motion signal", then what is the unit? How many flies are summed up?

Discussion is missing on the evidentiary value of the neurogenetic experiments. Kir/+ controls are vastly different in 5A and 5B. Why that discrepancy? Suppl.S5 Kir/+ control looks like 5B, not like 5A (same data set as 5B?). The split-GAL4 line shown in Fig.5 does contain LC-11 as one component. Then there might still be an expression outside LC11 causing the changes in behaviour.

Minor

Method: The description of the setup does not say whether the flies can walk under the ceiling of the chamber (could be prevented by coating). The behaviour is different when the flies' actions are restricted to the floor.

Error bars in all figures and suppl. figures are not defined in the legends.

Fig.3 D left/right. I suggest using an additional letter E.

L.30 "...wide prevalence, the..." (comma)

L.358 to stay frozen

L.368 variables β_i and β_s are not defined.

Letter to reviewers

We would like to thank the reviewers for their time and useful comments that improved our article. We have performed new experiments and new analysis, which we believe strengthen our claims. In the next three paragraphs we briefly address comments made by multiple referees. After these you can find our point-by-point answers to the reviewers' comments. We hope you find our responses satisfactory.

One point raised by multiple reviewers related to the step change between groups smaller than 5 flies and groups bigger than 5, one suggestion being that fly density might contribute to this observation. We have added a new experiment addressing this issue, that suggests that density is unlikely to explain the difference between the smaller and the bigger groups. Still density is likely to contribute to the social regulation of freezing, something that we are still exploring in the lab but for which new set-ups and experimental protocols need to be developed.

Another point raised by multiple reviewers regards the contribution of sensory modalities other than vision. As suggested by the reviewers we tested several mutants affecting olfactory, gustatory and mechanosensory modalities. We found that olfaction is unlikely to play a role in social regulation of freezing, as olfactory mutants froze when tested individually and less so when tested in groups. However, the other two mutants, affecting gustation and mechanosensation did not yield conclusive results as both affected freezing even in flies tested individually. Still we decided to further explore the role of contacts by adding those as a parameter in our model and found that contact does not increase the explanatory power of our model. Hence, gustation and mechanosensation are unlikely to contribute to the observed decrease in freezing of flies tested in groups.

Finally, reviewers raised concerns regarding the parental controls we used for the neuronal silencing experiments. To address these, we repeated our experiments now comparing the experimental lines with either Empty GAL4 or Empty split-GAL4 as controls. In addition, we used another neuronal silencer for which there is an inactive form which we believe is the most complete control (we used active-TNT and its inactive form as control). When running these experiments, the split-GAL4 lines gave mixed unreliable results. We would need to order new stocks and perform all experiments to be confident regarding this line. One issue is that split lines have often weaker expression levels and thus lead to less robust effects. At this stage given the extraordinary circumstances we are not in a position to start these experiments. Therefore, we chose to report only the results regarding the GAL4 line which we fully trust. Although the GAL4 line is quite sparse it does label a few neurons other than LC11. These additional neurons are not, to our knowledge involved in visual processing. Since we found that visual motion is a very strong modulator of freezing, we believe the effects we observed are most likely mediated by LC11 neurons. we altered the text to acknowledge that other neurons targeted in this line may contribute to the effects observed.

Reviewer #1:

1. Figure 1B: the grouping of behavioral responses into three categories (i.e., 1 fly, 2-5 flies and 6-10 flies), as opposed to a linear impact of group size (animal density) is intriguing. Can the authors comment on what might define the apparently threshold between groups of 5 flies and groups of 6 flies?

We thank the reviewer for the pertinent remark regarding the stark change in effect size for groups bigger than 5. Reviewer #2 raised a similar question, suggesting that it may relate to fly

density. Indeed, our lab is very interested in exploring the effect of population density and spatial structure on the group behaviour. This, however, implies developing a new set-up that allows studying much larger groups in much larger arenas, which we believe is out of the scope of the current manuscript. Still, we do agree with both reviewers that the step between 5 and 6 flies is intriguing and might relate to fly density. Therefore, as an initial approach we tested groups of 5 flies in smaller arenas effectively increasing fly density closer to that observed in the original arenas with 7 flies. More specifically, we compared the behaviour of groups of 5 flies in our standard arenas to that in arenas that are 1 cm smaller, changing the density from 0.14 fly/cm to 0.19 fly/cm (fly density of groups of 7 in the standard arena was also 0.19 fly/cm). We found that the behaviour of 5 flies in the two arenas was similar, suggesting that density is unlikely to explain the steep decrease in freezing between groups of 5 flies and higher numbers. We have added a new supplementary figure 2 (see below) showing this data and added the following to lines 71-74 in the article's main text:

'The step-wise decrease between groups of 5 and 6 flies, does not seem to depend on fly density, as testing groups of 5 flies in a chamber that is 1 cm smaller, creating a density similar to that in groups of 7, did not impact freezing responses (Supplemental Fig. 2).'

Fig. S2. Density effect on freezing behaviour in groups of 5 flies. A) Proportion of flies freezing throughout the experiment, for groups of 5 flies in 58 mm diameter arenas or in 68 mm diameter arenas (standard arena). B–D) Violin plots representing the probability density of individual fly data bound to the range of possible values, with boxplots. B) Proportion of time spent freezing throughout the experiment. C) Probability of freezing entry in the 500 ms bin following looming presentation. D) Probability of freezing exit in the 500 ms bin before the following looming stimulus. P-values result from two-tailed Mann-Whitney test.

2. Line 90: typo in "were"

'Where' was changed to 'were'.

3. *Figure 2A: I don't understand the color key; in the plot there are orange bars and dark orange bars, but the key indicates orange, black and grey.*

We thank the reviewer for this comment which improves the readability of Figure 2A. This mismatch in colours was due to the juxtaposition of the colours using transparency. To make this clearer and more intuitive, we have changed the legend to use dark orange instead of black. As for the grey, symbolizing looming offset in both graphs, we changed the full line to a thinner dashed one.

4. *Figure 5B: the Kir2.1 control parental line show higher freezing behavior than in panel A (and the control Gal4 line). Is this just a fluke of this set of experiments?*

We thank the reviewer for this observation. Indeed, the parental Kir2.1. control shows higher freezing in B compared to A. We do observe variability in behaviour across experiments performed in different days, which we account for as each experimental line is run with interspersed controls. Therefore, comparisons between experimental lines and controls are sound, but comparisons across experiments are difficult to interpret. Still, to confirm our results we repeated the experiment with Kir2.1 now comparing with an Empty-GAL4 line instead of the parental controls. We found the same result, which is now in the revised version of Figure 5 (original plot with parental controls are in revised Supplementary Figure 6).

5. *The use of auditory cues as defense signals in other species made me wonder whether this sensory modality contributes to social regulation of freezing behavior in flies. (As a precedent, analysis of courtship has shown a male responds to auditory cues from the footsteps of females (PMID: 18802468)). While visual cues are most likely to be the most important signal, can the authors rule in or out other modalities? (No experiments are needed, but they could comment on this in the Discussion).*

We thank the reviewer for this pertinent comment, which taps into an area we have explored a bit. Our experiments with the magnets, which do not contain any of the fly chemosensory signals, show that neither olfaction nor taste are required for the group effect on freezing onset. On the other hand, we have explored the role of other sensory systems, by testing olfactory mutants (near-anosmic mutants IR8a¹, IR25a², GR63a¹, ORCO¹), gustatory mutants (PoxNeuro) and mechanosensory mutants (*inactive*). This preliminary work showed that: olfaction did not play a prominent role as flies both individually or in groups froze to levels similar to that controls (CS); mechanosensory mutants (*inactive*), which affect chordotonal organs and therefore both audition and proprioception did not display freezing responses in individuals or in groups, suggesting a role for mechanosensation in freezing independently of the social environment (this is now an active research avenue in the lab); finally, gustatory mutants (PoxNeuro) showed motor impairments whether tested alone or in groups, which impacted freezing responses in both cases.

To address this issue, we have added a supplementary figure (see below), where we present the olfactory data, and also the impact of adding collisions to our model. The fact that adding contacts to our model does not improve the model's predictive capability, argues that gustation

are unlikely to play a major role in the group response. We also altered the text in the results section, lines 189-192, to read:

“Finally, we also assessed the role of other sensory cues, namely olfaction and gustation. Using near-anosmic mutants and testing the impact of contacts, required for gustatory cues, on the logistic regression model we found that olfaction and gustation are unlikely to play a role in the group response (Supplemental Fig. 6).”

As for mechanosensation, we have further added the following to the discussion:

“Though motion also generates vibrations cues and these can be used to detect the movement of other flies ³⁵, these results suggest visual cues play a predominant role.”

Fig. S6. The role of other sensory modalities on social regulation of freezing responses in groups. A) Proportion of flies freezing throughout the experiment for flies tested individually and in groups of 5 wild-type flies and near-anosmic olfactory mutants. B–D) Violin plots representing the probability density of individual fly data bound to the range of possible values, with boxplots. B) Proportion of time spent freezing throughout the experiment. C) Probability of freezing entry in the 500 ms bin following looming presentation. D) Probability of freezing exit in the 500 ms bin before the following looming stimulus. P-values result from Kruskal-Wallis statistical analysis followed by Dunn’s multiple comparisons test. E) Adding collisions to the logistic regression model presented in Figure 3, does not improve the model’s predictive capability; the fact that contacts do not seem to be important for the group response argues against a role for gustation in this process. Mechanosensation may still play a role but see discussion section.

6. Discussion: this feels a bit long - the conclusions and broader implications of the work would come across stronger with a bit of pruning. For example, lines 216-231 are more background material (which might fit better in the (currently brief) introduction), and in lines 257-267, the highlighting of potential parallels between fly and visual processing feels a bit labored (and not really key for this paper).

We thank the reviewer for this comment, which we feel increases the readability of the paper, creating a better narrative. We have moved previous lines 216-231 to the introductions, lines

31-43, and we have taken out the parallels with retinal ganglial cells, such that now the parallels with the vertebrate visual processing system is restricted to the following:

‘The parallels between visual systems of flies and humans (reviewed in ^{34,35}), despite the lack of any common ancestor with an image forming visual system, suggest that shared mechanisms underlying visuomotor transformations represent general solutions to common problems that all organisms face individually or as a group.’

7. Line 320: the web-link given is uninformative (I was expecting to have access the software).

We are in the process of upgrading the way we analyse our data and will soon make the code available. As such, we have taken out the web-link as some of the features of the now outdated software we do not use and instead calculate them as per the methods section.

Reviewer #2:

Major points:

1. *In figure 1, authors showed that flies in groups of 6 to 10 were less likely to enter to a freezing state compared to individuals and groups of 2 to 5 flies. The difference between groups of 2-5 and groups of 6-10 is a step wise increase rather than a gradual difference. Is this an artifact of the chamber size or is this a biologically relevant result? If authors increase the size of the chamber would they expect different results in their assays?*

We thank the reviewer for this pertinent point regarding the step change in the effect of group size. Please see also our answer to reviewer#1. Briefly we added an experiment where groups of 5 flies were tested in a smaller arena, such that the density was closer to that of groups of 7, and compared that to groups of 5 in the original arena. We found no difference between the two conditions, suggesting that density is unlikely to explain the big drop in freezing when we test groups of more than 5 flies. We have added a new Supplementary Figure 2 showing this data and added the following to the article’s main text:

‘The step-wise decrease between groups of 5 and 6 flies, does not seem to depend on fly density, as testing groups of 5 flies in a chamber that is a 1 cm smaller, creating a density similar to that in groups of 7, did not impact freezing responses (Supplemental Fig. 2).’

2. *In supplementary figure 2, authors investigate whether mechanosensory interactions contribute to freezing behavior and found minor impact of collisions on freezing behavior. Based on these preliminary tests, authors conclude that freezing behavior is not related to mechanosensation. I think these experiments are a good start however they need to be expanded to exclude the possibility that any proprioceptive feedback is not involved in this behavior. Authors also ignore, any chemosensory signal that can impact this behavior. In many species, alarm pheromones play a key role in directing group behavior to predators or threats in the environment. These experiments are simple, fly mutants are available to test impact of olfactory or taste signal on freezing behavior in fly groups. I think authors either need to conduct these experiments or explain in the text why*

they exclude the possibility that chemosensation or proprioception can regulate freezing behavior.

We thank the reviewer for this comment. We do not wish to claim that vision is the sole mediator of the social regulation of freezing breaks, but rather that in this context it is the main sensory modality responsible for the effect. In addition, in this study we are not making claims regarding the sensory modalities involved in freezing behaviour itself, but rather the social regulation of freezing. Regarding proprioceptive feedback, we do indeed believe that it is important for freezing, constituting an active research avenue in the lab. We tested *inactive* mutants and found that these mutants did not freeze (see reply to reviewer 1). In our reply to reviewer 1 we also address other sensory modalities such as olfaction and gustation (see above).

Here we would like to address further the role of mechanosensation. Since using mutants does not allow us to test the role of mechanosensation in the social regulation of freezing, we further assessed the role of collisions whose effect would rely at least in part on this sensory modality. To this end, we added contacts to our logistic regression model used to predict freezing breaks. Contacts were added as a variable that was either 1 or 0, i.e., we took into account whether there had been at least one collision or none during each type of freezing bout (with or without a freezing break before the next loom). As displayed below this did not improve the model's predictive capability, nor does this variable explain our data. We have added this figure to supplemental Figure 6.

3. In figure 5, authors showed that inhibiting LC11 neurons by Kir2.1 expression reduces group effect on looming-triggered freezing behavior and they argue that LC11 neurons are involved in processing motion cues of other flies to regulate freezing response. In these experiments Kir2.1 is expressed throughout the development and LC11 neurons have been constantly hyperpolarized. These experiments need to be repeated by silencing LC11 neurons only in adults using temporal inactivation of these neurons with available reagents such as temperature controlled shibire. In addition, they can also activate LC11 neurons and examine whether freezing behavior will be suppressed in individual flies when LC11 neurons are active during looming stimulation. If LC11 neurons indeed mediate freezing responses in grouped flies, activation of these neurons should induce the opposite phenotype caused by neural silencing.

We thank the reviewer for pointing out the possible role of developmental effects, which we agree is important to address.

We decided to use Tetanus Toxin (TNT), as we had both a line with constitutive and inducible TNT (under the control of temperature sensitive GAL80^{ts}, a GAL4 repressor). For these experiments, we first compared the constitutive expression of an active form of TNT with that of an inactive form. We found that constitutive expression of active TNT yields a similar result to the expression of Kir2.1, that is, expressing TNT driven by LC11-GAL4 resulted in increased freezing of flies tested in groups of 5 as compared to groups of 5 control flies that expressed the inactive form of TNT. We now added the results from this experiment to the revised figure 5 (see plot below).

We then tested the effect of conditional expression of TNT for which we used 5xUAS-TNT, tubulin-GAL80^{ts} driven by LC11-GAL4. In this case as control we used Empty-GAL4, as we did not have the inactive form of TNT under the control of GAL80^{ts}. Flies were grown at 18° the permissive temperature for GAL80^{ts}. LC11GAL4>TNT were split into two conditions, one kept at 18° and the other was shifted to 30° prior to the behavioural experiments. Both cases were placed at 25° for 4h prior to the experiment, the temperature at which exposure to looming was performed. The same two conditions were applied to control Empty-GAL4 flies. In these experimental conditions driving 5xUAS-TNT, tubulin-Gal80^{ts} in LC11-GAL4 failed to replicate the phenotype obtained with constitutive expression of TNT.

Unfortunately, the lines available for the constitutive and conditional inactivation experiments were not the same. At this stage it is possible that the lack of effect of the conditional expression of TNT is due to the lines used (we don't have any other experiments in the lab with these lines to confirm they are working), or because the effects we observed with constitutive inactivation are developmental in nature. Therefore, we cannot at this moment determine with certainty

whether the effect we observe is developmental or adult specific. In order to unambiguously answer this question, we would have to first generate these lines, which would require additional time. We have therefore added the following to lines 219-221:

‘In addition, the observed effect of silencing neurons targeted by the LC11-GAL4 line on freezing in groups may be adult specific or due to developmental effects.’

The reviewer also suggests an experiment where we optogenetically activate LC11 neurons in an attempt to mimic the social regulation of freezing. The prediction being that activating these neurons should make flies more likely to break from freezing. These experiments although interesting are only conclusive with a positive effect, since artificially activating neurons may not recapitulate the natural neuronal activity patterns and therefore fail to mimic natural behaviour, even when natural activity of these targeted neurons might be sufficient to mimic behaviour. Still, we tried to activate LC11 neurons in two different ways. We first tried with optogenetic activation for its temporal precision. However, when piloting the activation protocol as flies are exposed to looming stimuli, we found that the red light itself at the intensities required to activate chrimson, a red-shifted channel rhodopsin, disrupted the flies’ behaviour. Next we tried to activate neurons through constitutive expression of NaChBac, which did not affect the behaviour. Multiple reasons could explain the lack of effect including that NaChBac was not depolarizing cells enough to result in a behavioral effect. Again, we do not have in the lab experiments that suggest the line we used is working. The usage of temperature sensitive approaches that require a temperature shift during the expression of the behaviour do not work for our behaviour as they affect locomotion, which in turn affects freezing propensity (as our group showed in Zacarias et al., 2019).

4. In figure 5B, there is a significant difference between two control groups in freezing behavior; LC11 split-GAL4 and UAS-Kir2.1. What is the source of this variation in freezing behavior? Did they observe any behavioral or genetic problems in the LC11 split-GAL4 strain? Please clarify.

We thank the reviewer for pointing out this difference. We did not observe any obvious behavioural or genetic problems with the LC11 split-GAL4 line. We did however observe some variability in the UAS-Kir2.1 parental control across our experiments. We have now repeated experiments where the experimental line is compared to Empty-GAL4 or Empty-Split-GAL4, as well as added another silencer for which there is an inactive form which we believe is the most complete control (we used active-TNT and its inactive form as control, see above). As explained at the very beginning of this letter we have removed from the manuscript the split-GAL4 experiments, that we found yielded mixed results, and altered the manuscript to acknowledge the possible role of other neurons that are targeted in the LC11-GAL4 line.

We have added the following to lines 215-219:

‘These data, together with the identification of visual motion cues as mediators of group freezing responses, point to the role of LC11 neurons in this process. However, given that *LC11-GAL4*, despite its sparseness, also directs expression outside these neurons, namely in the descending neurons DNg26³², we cannot at this moment fully rule out the effect of expression outside LC11.’

In addition, the observed effect of silencing neurons targeted by the LC11-GAL4 line on freezing in groups may be adult specific or due to developmental effects.

Minor points:

1. Please revise the gray color scheme in Figure 2. It is very hard to distinguish different experimental groups especially in Figure 2C. Also, please indicate the control groups in the figures more clearly. Stopped magnets and freezing magnets sounds too similar and confusing for the reader.

In order to increase clarity we have added the following to the figure legend:

‘Simulating groups of 5 using movable magnets (stopped magnets – immobile magnets throughout the experiment; magnets freezing – magnets move during baseline becoming immobile from the onset of the stimulation period)’.

2. In the line 279, authors mentioned that they used mated females for their experiments. Mating could influence various aspects of animal behaviors including foraging, egg laying, courtship, and receptivity. Would they expect differences in freezing behavior if virgin females or males are used?

We thank the reviewer for this comment, which taps into our current and future research endeavours focused on factors that modulate this group response. We are currently looking at sex differences, and indeed males seem to behave slightly differently, in that they have a stronger response to the group. We are exploring the reasons for this difference. As for virgin females, we have not started these experiments, but we are interested in exploring them in the future. We are not yet ready to comment on this issue in this paper.

3. Authors use blind flies in the Figure 4, but they do not describe the genotype or the strain used in the methods. Please indicate which blind fly strain is used in the paper.

We list the genotype (NorpA mutant flies) in the methods but have also added it into the main text for clarity (line 169).

Reviewer #3:

Major

1. Fig.3 B right side x-y-plots. Y-axis labels are missing and plots are not explained in the legend. What is the difference (despite x-axis range)? Shouldn't the 3rd contain the 2nd from -4s to 0s and the 2nd the first from -0.6s to 0s?. If this is "Summed motion signal", then what is the unit? How many flies are summed up?

We thank the reviewer for pointing out the issues with the plot. We have added the Y-axis legend (motion signal), as well as an X-axis for the heat maps. Each heatmap and plot pair

correspond to a single representative event, where the heatmap represents the motion signal for each of the surrounding four flies and the line plot represents the summed motion signal of those same four flies. In order to attempt to make this clearer we have changed the legend from:

'B) The motion signal is formalized as the other fly's speed multiplied by the angle it produces on the retina of the focal fly (schematic). Representative examples of the motion signal starting in the 500 ms bin after looming offset by a focal fly until freezing exit or the end of the inter-loom interval (without freezing exit): heatmaps show the motion signals of the 4 surrounding flies and the line graphs show the summed motion signal.'

To:

'B) The motion signal is formalized as the other fly's speed multiplied by the angle it produces on the retina of the focal fly (schematic). Representative examples of the motion signal starting in the 500 ms bin after looming offset for a focal fly until freezing exit or the end of the inter-loom interval (without freezing exit): heatmaps show the individual motion signals for each of the 4 surrounding flies and the line graphs show the summed motion signal of these 4 flies.'

2. Discussion is missing on the evidentiary value of the neurogenetic experiments. Kir/+ controls are vastly different in 5A and 5B. Why that discrepancy? Suppl.S5 Kir/+ control looks like 5B, not like 5A (same data set as 5B?). The split-GAL4 line shown in Fig.5 does contain LC-11 as one component. Then there might still be an expression outside LC11 causing the changes in behaviour.

We thank the reviewer for this comment. The controls present in the datasets in Figs 5B and Suppl. S5 are different. Indeed, we do see a difference in Kir/+ controls in figure 5B. As mentioned in our answers above, we have repeated the experiments using more appropriate controls, Empty-GAL4. In addition, we also silenced neurons using TNT and used the inactive form of TNT as control. As for expression outside LC11, indeed we cannot rule out the contribution of other neurons also targeted in the lines we used. Thus, we have added the following sentence (lines xxx) which addresses the issue of the expression outside LC11 in the L11-Gal4 line:

'These data, together with the identification of visual motion cues as mediators of group freezing responses, point to the role of LC11 neurons in this process. However, given that *LC11-GAL4*, despite its sparseness, also directs expression outside these neurons, namely in the descending neurons DNg26³², we cannot at this moment fully rule out the effect of expression outside LC11.'

Minor

3. Method: The description of the setup does not say whether the flies can walk under the ceiling of the chamber (could be prevented by coating). The behaviour is different when the flies' actions are restricted to the floor.

We did not restrict the flies to the floor as preliminary data using sigmacote to prevent this did not show substantial differences. We also analysed freezing responses for flies that were in the ceiling or on the floor at the time of the looming stimulus, and found no difference.

We have added the following to the setup description in the methods section:

'Flies were not restricted to the arena floor, as during initial experiments we observed no difference in defensive responses for flies on the floor or ceiling.'

4. Error bars in all figures and suppl. figures are not defined in the legends.

We thank the reviewer for this comment. The violin plots do not contain error bars, but rather represent the entire data and contain standard boxplot representation with median and interquartile range. The line that extends from the boxplot represents the minimum and the maximum value. Each figure legend has the description of the graph:

"Violin plots representing the probability density of individual fly data bound to the range of possible values, with boxplots"

5. Fig.3 D left/right. I suggest using an additional letter E.

We changed the figure accordingly.

6. L.30 "...wide prevalence, the..." (comma)

We added the comma.

7. L.358 to stay frozen

We changed 'to stay freezing' to 'to stay frozen'.

8. L.368 variables β_i and β_s are not defined.

We added the following 'where β_s and β_i are, respectively, the β -coefficients of social cues and individual behavior.'

Reviewers' Comments:

Reviewer #1:

Remarks to the Author:

This is a strong review that addresses both my and the other reviewers' comments adequately, through new experiments and some wise toning down of conclusions where certain experimental results have (unavoidable) caveats.

Reviewer #2:

Remarks to the Author:

In this revised version of the manuscript by by Ferreira and Moita, authors have added useful behavioral experiments that clarified most of the confusion I have with the paper. They have also significantly improved the description of the behavior experiments, and provided necessary controls. After the substantial changes done in the text, I think the data provided supports the conclusions of the paper. I thank the authors for their hard work. I have no further comments, and I recommend publication of the manuscript with no further edits.

Reviewer #3:

Remarks to the Author:

Thank you for the revisions, which addressed most of my points. Two minor points remain.

It is still unclear, what boxplots within your violin plots represent. Whiskers of boxplots may represent 1.5 IQR, or 10%/90%, or 5%/95%. In your rebuttal letter you say IQR, but this information does not appear in the legends of the manuscript.

New point:

Please reconsider Line 255 in the discussion section saying "insects can only use short range auditory signals". This is certainly not true for crickets, grasshoppers and other insect auditory-communication specialists. Their auditory signaling range goes far beyond their visual range.

REVIEWERS' COMMENTS:

Reviewer #1 and Reviewer #2 required no answer.

Reviewer #3 (Remarks to the Author):

Thank you for the revisions, which addressed most of my points. Two minor points remain.

It is still unclear, what boxplots within your violin plots represent. Whiskers of boxplots may represent 1.5 IQR, or 10%/90%, or 5%/95%. In your rebuttal letter you say IQR, but this information does not appear in the legends of the manuscript.

We thank the reviewer for this comment. The whiskers represent the $1.5 \times$ IQR. We have added the following description of the boxplot elements to the figure legends: '(elements: center line, median; box limits, upper (75) and lower (25) quartiles; whiskers, 1.5x interquartile range).'

New point:

Please reconsider Line 255 in the discussion section saying "insects can only use short range auditory signals". This is certainly not true for crickets, grasshoppers and other insect auditory-communication specialists. Their auditory signaling range goes far beyond their visual range.

We thank the reviewer for pointing out this inaccuracy; indeed we had *Drosophila* in mind and so we have changed the line to read: "*Drosophila melanogaster* uses short range auditory signals"